

# Joint embedding VQA model based on dynamic word vector

Zhiyang Ma[1], Wenfeng Zheng[1], Xiaobing Chen[1] and Lirong Yin[2]

[1] School of Automation, University of Electronic Science and Technology of China, Chengdu, P. R. China
[2] Department of Geography and Anthropology, Louisiana State University, LA, USA

## ABSTRACT

The existing joint embedding Visual Question Answering models use different combinations of image characterization, text characterization and feature fusion method, but all the existing models use static word vectors for text characterization. However, in the real language environment, the same word may represent different meanings in different contexts, and may also be used as different grammatical components. These differences cannot be effectively expressed by static word vectors, so there may be semantic and grammatical deviations. In order to solve this problem, our article constructs a joint embedding model based on dynamic word vector—none KB-Specific network (N-KBSN) model which is different from commonly used Visual Question Answering models based on static word vectors. The N-KBSN model consists of three main parts: question text and image feature extraction module, self attention and guided attention module, feature fusion and classifier module. Among them, the key parts of N-KBSN model are: image characterization based on Faster R-CNN, text characterization based on ELMo and feature enhancement based on multi-head attention mechanism. The experimental results show that the N-KBSN constructed in our experiment is better than the other 2017—winner (glove) model and 2019—winner (glove) model. The introduction of dynamic word vector improves the accuracy of the overall results.

## INTRODUCTION

The Visual Question Answering (VQA) system takes a picture, and a free and open natural language question about this picture as input, and generates a natural language answer as output. VQA has many potential applications. The VQA system can be integrated with other research areas, such as social media, e-commerce, medical Science (*Abacha et al. 2019*; *Tang et al., 2020*), seismology (*Potnis, Shinde & Durbha, 2021*; *Yin et al., 2019*; *Zheng et al., 2016*), disabilities (*Gelšvartas, Simutis & Maskeliūnas, 2016*), gaming (*Atkinson et al., 2019*) and chatbots (*Klopfenstein, Delpriori & Ricci, 2018*). Since the appearance of visual question answering task in 2015, a large number of VQA model belong to the joint embedding model. Firstly, visual information and question text information are respectively characterized in the joint embedding model, and then image features and text features are fused and finally predict the answer through the classifier. Because the model of this architecture is easy to be trained, researchers have tried a lot of

Corresponding author
Wenfeng Zheng,
wenfeng.zheng.cn@gmail.com

different image feature extraction methods, different text feature extraction methods, and different fusion methods of the two modals.

In the aspect of text characterization, *Zhou et al. (2015)* proposed the iBOWIMG model, and transferred the pre-trained GoogleNet (*Szegedy et al., 2015*) to extract image features. *Gao et al. (2015)* thought that the question and answer are different in syntactic structure, so they used two independent LSTM networks to encode the question and decode the answer, and combined the convolutional neural network to form the mQA model. Lin et al. proposed a dual CNN model. They applied the convolutional neural network to both the encoding of the image and the feature extraction of the question text, and used a multimodal convolution layer to output the joint eigenvectors (*Ma, Lu & Li, 2015*).

In the aspect of image feature extraction, in addition to different models using different pre-trained CNN, *Noh, Hongsuck Seo & Han (2016)* thought that the deep convolution neural network with single weight configuration can not effectively deal with different problems. They added a dynamic parameter layer to the convolutional neural network.

In addition to using different methods to extract image and text features, cross modal feature fusion method has been widely studied. *Malinowski, Rohrbach & Fritz (2015)* proved that the accuracy of the system is related to the feature vector fusion method by comparing different feature vector fusion methods. *Antol et al. (2015)* used the element by element multiplication feature fusion method. *Fukui et al. (2016)* thought that in the external multiplication operation between vectors, the interaction between all elements was more active and can retain more abundant feature information, so a more complex multimodal compact bilinear pooling method (MCB) was proposed. *Saito et al. (2017)* thought that different feature fusion methods would retain different levels of features. Therefore, the model Dualnet which combines element by element addition and element by element multiplication is proposed.

Attention mechanism has been proved to be effective in a large number of deep learning tasks, attention mechanism is also widely used in VQA model. *Chen et al. (2015)* firstly introduced attention mechanism into visual question answering task, and proposed a configurable convolutional neural network (ABC-CNN) based on attention mechanism to reduce the influence of irrelevant regions. The test results on Toronto COCO-QA (*Ren, Kiros & Zemel, 2015*), DAQUAR (*Malinowski & Fritz, 2014*) and VQA (*Antol et al., 2015*) all achieved optimal results, which proved the effectiveness of attention mechanism in improving VQA tasks.

This article analyzes the following reasons for the excellent performance of joint embedding model in VQA challenge:

1. Attention mechanism is introduced. In 2016, the winning model (*Ilievski, Yan & Feng, 2016*) proposed the dynamic attention (FDA) model. The purpose is to dynamically allocate weights to different regions of the image according to the keywords in the question, so as to obtain the combination of global features and local features of the image. The top-down and bottom-up image attention mechanisms of article (*Anderson et al., 2018*) were used in the winning models in 2017 and 2018, and the multi-head

attention mechanism of Transformer (*Vaswani et al., 2017*) was used in the winning model in 2019. The introduction of attention mechanism can reduce the interference of irrelevant features, improve the computational efficiency, and improve the interpretability to a certain extent

2. The limitations of VQA2.0 dataset. VQA challenge takes VQA2.0 as the data set. However only 5.5% of all questions require common sense or external knowledge in VQA2.0, which means that no additional information is needed to answer most of the questions (*Wang et al., 2015*). However, in the open questions in reality, questions involving common sense or external knowledge exist widely. Therefore, VQA2.0 dataset has limitations, which makes the model only need to focus on images and text, so joint embedding model has become the main architecture.

3. Thanks to the progress of image recognition and natural language processing model. The joint embedding model has flexible combination pattern, and it is easy to transfer the model which performs well in other tasks to the forming of a new model.

As shown in the Table 1, the existing joint embedding models (*Anderson et al., 2018*; *Antol et al., 2015*; *Fukui et al., 2016*; *Ma, Lu & Li, 2015*; *Noh, Hongsuck Seo & Han, 2016*; *Teney et al., 2018*; *Yu et al., 2019*; *Zhou et al., 2015*) use different combinations of image characterization, text characterization and feature fusion methods, but all existing models use static word vectors for text characterization. The static word vector uses a corpus as the data set to train and get the distributed representation of each word. The advantage of this representation method is that the word vectors are obtained through pre-training. Therefore, when it is applied to different downstream tasks, there is no need to retrain and the computational efficiency is improved. However, in the real language environment, the same word may represent different meanings in different contexts, and may also be used as different grammatical components. These differences cannot be effectively expressed by static word vectors, so there may be semantic and grammatical deviations.

In order to solve the problem of static word vector, our experiment constructs a joint embedding model based on dynamic word vector—none KB-Specific network (N-KBSN) model. Our article will focus on the N-KBSN model, and use VQA2.0 dataset for training. N-KBSN consists of three main parts: question text and image feature extraction module, self attention and guided attention module, feature fusion and classifier. Faster R-CNN, which is excellent in multi-label detection, is used for image feature extraction. According to *Peters et al. (2018)*, the ELMo representations ideally model both (1) complex characteristics of word use (e.g., syntax and semantics), and (2) how these uses vary across linguistic contexts (i.e., to model polysemy). And ELMo representations can be easily added to existing models and significantly improve the state of challenging NLP problems. Inspired by this, in our VQA experiment, ELMo model which can obtain context information is used for feature extraction of question text, and multi-head attention mechanism (*Vaswani et al., 2017*) learned from Transformer is used to realize image self attention (V-SA), question text self attention (Q-SA) and image attention guided by the question (Guided Attention, GA), and the answer is predicted by feature fusion. The infrastructure of N-KBSN model is shown in Fig. 1.

**Table 1 Comparison of representative joint embedding models.**

| Model | Image characterization | Text characterization | Feature fusion | VQA accuracy(%) | Static word vector |
|---|---|---|---|---|---|
| LSTMQ+I | VGGnet | LSTM | Itemized multiplication | 54.1 | Yes |
| iBOWIMG | GoogleNet | BoW | Series connection | 55.9 | Yes |
| DPPNet | VGGnet | GRU | Dynamic parameter layer | 57.4 | Yes |
| d-cnn | CNN | CNN | CNN | 58.4 | Yes |
| MCB | RestNet | LSTM | MCB | 64.2 | Yes |
| 2017-winer | Faster R-CNN | Glove+GRU | Itemized multiplication | 69.87 | Yes |
| 2018-winer | Faster R-CNN | Glove+GRU | Itemized multiplication | 72.27 | Yes |
| 2019-winer | Faster R-CNN | Glove+LSTM | MLP | 75.26 | Yes |

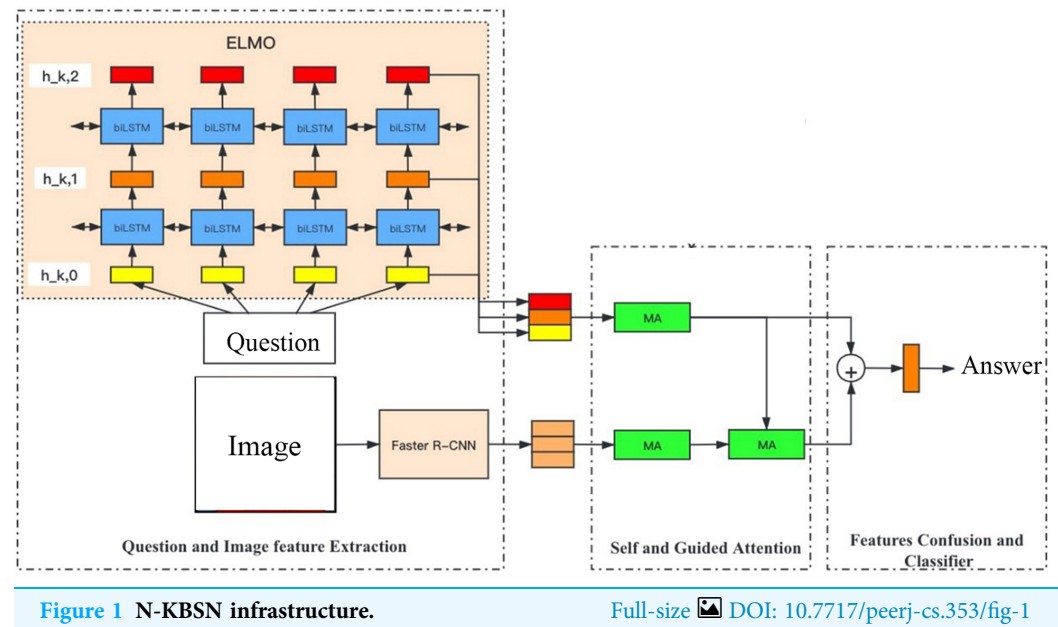

**Figure 1 N-KBSN infrastructure.**

# DATA

Since 2014, several high-quality VQA data sets have been proposed: DAQUAR (*Malinowski & Fritz, 2014*), COCO-QA (*Ren, Kiros & Zemel, 2015*), VQA (*Antol et al., 2015*), VQA 2.0 (*Goyal et al., 2017*), CLEVR (*Johnson et al., 2017*), KB-VQA (*Wang et al., 2015*), FVQA (*Wang et al., 2017*). Among which, the KB-VQA and FVQA are knowledge-based dataset.

1. DAQUAR dataset contains 1,449 images, most of which are indoor scenes, which greatly limits the richness of the scene of the dataset, which is a major disadvantage of the dataset. The data set consists of training set and test set. The question and answer pairs are generated by algorithms or provided by human volunteers according to the given template.

2. COCO-QA contains 123,287 real scene images from MS COCO (*Lin et al., 2014*). The questions are divided into four types: object recognition, color recognition,

counting and location query. In the actual test process, DAQUAR data set has been found to be able to obtain high accuracy only by simply guessing the answer, which makes the high accuracy rate appear a great deviation, which cannot fairly test the reasoning ability of the system. In order to overcome this shortcoming, COCO-QA eliminates some answers with extremely low and extremely high frequency, making the frequency of common answers decrease from 24.98% to 7.30%.

3. VQA data set is an important turning point in the development of visual question answering field. Before that, the question types of previous data sets were limited to some templates, which made the data sets unable to test the performance of visual question answering system in real context. For example, DAQUAR limited the answers to 16 basic colors and 894 object categories. The questions and answers in VQA dataset are unlimited and open, and all of them are generated by human beings. At the same time, the number of pictures is two orders of magnitude higher than DAQUAR, reaching 254,731, which greatly improves the capacity of the data set. The VQA dataset contains not only the 204,721 images of real scenes and 50,000 synthetic abstract scene images are provided, which enriches the diversity of database scenes and facilitates high-order scene reasoning and complex spatial reasoning.

4. VQA 2.0. Aimed at the question of language bias in VQA dataset, VQA 2.0 enhances the visual information of VQA dataset by adding new confused data to the original VQA dataset. Confused data is organized in the form of (image I, question Q, answer A) just like the original data. The difference is that the newly added image is similar to the original image, but the answer to the same question Q gives a different answer A. For the same question, facing different pictures, we need to get different answers, which requires that the system not only can understand the natural language questions, but also can accurately identify the differences between pictures. This balanced method can filter out the algorithms that weaken image understanding, and enhance the importance of image understanding in VQA tasks. The supplementary VQA 2.0 contains 1.1 million pairs of image-question and 200,000 real scene images related with 1,300 million questions. The data volume is almost twice as much as that of VQA data set, and has become a new test standard for open questions.

5. CLEVR. In order to measure the reasoning ability of visual question answering system more accurately, *Johnson et al. (2017)* proposed CLEVR data set which combines language and basic visual reasoning.

In this experiment, it is enough to use VQA2.0 data set to train visual question answering model. The specific information about the data set is introduced in "Experimental Setup".

# METHOD AND EXPERIMENTS

## Image characterization based on FasterR-CNN

Target detection aims at locating the target accurately and quickly from the image, and recognizing the characteristics of the object such as its category. Traditional target detection algorithms select regions, extract features and classify them by sliding window,

such as deformable components Model (DPM) method (*Felzenszwalb et al., 2009*). Most of these traditional target detection algorithms have poor performance and high time complexity. Deep learning has been performing well in VQA tasks and there are a large number of excellent target detection algorithms, such as R-CNN (*Girshick et al., 2014*), SPP-Net (*He et al., 2015*), Fast R-CNN (*Girshick, 2015*), Faster R-CNN (*Ren et al., 2015*), Mask R-CNN (*He et al., 2017*), YOLO (*Redmon et al., 2016*) and so on. Because Faster R-CNN has shown excellent performance in various target recognition tasks, our experiment uses Faster R-CNN to extract image features.

FasterR-CNN is divided into four modules: convolution layer, regional proposal network (RPN), ROI pooling and classifier (*Ren et al., 2015*). CNN and its variants are used by the convolution layer to extract image features. The RPN network is used to predict the candidate regions. The network slides on the image features output by CNN. The network predicts the category score in each spatial region. The feature map and candidate region of the image are combined with the pooling layer of interested region to get the region feature map. Then, the fully-connected layer and classifier are used to predict the category of each region, and the detection frame of the object is obtained by candidate frame regression.

The N-KBSN model uses ResNet-101 which has been pr-trained on Imagenet (*Russakovsky et al., 2015*) and Faster R-CNN trained on Visual Genome (*Krishna et al., 2017*) to extract image features. Given the image I, the model extracts $m$ non-fixed size image features $X = \{x_1, x_2, ..., x_m\} x_i \in \mathbb{R}^D$, and each image feature encodes an image region. The feature dimension of each image region is 2,048. For the feature map output by the convolution layer, the model uses non-maximum suppression and Iou thresholds to select the top candidate regions. By setting a threshold of target detection probability, the network obtains a dynamic number of detected objects $m \in [10, 100]$, and uses zero padding to make $m = 100$. For each selected region I, $x_i$ is defined as the mean pooling result of the feature graph of the region, and $x_i$ of $m$ regions are spliced into the final image feature. Therefore, each input image will be transformed into a $100 \times 2,048$ image feature for subsequent attention module.

## Text characterization based on ELMo

As shown in Table 1, text characterization of previous visual question answering models is to obtain static word vectors through corpus learning, each word corresponds to a certain real number vector. This fixed vector does not perform well in dealing with the polysemy of words. Both Chinese and English words have the phenomenon of polysemy, the meaning of the same word changes in different contexts. In order to solve the problem of polysemy, dynamic word vector is proposed. And ELMo and BERT are the representatives. ELMo improves the accuracy of the model in multiple NLP tasks. Therefore, our experiment introduces ELMo model to deal with text in VQA tasks, and combines attention mechanism which is similar to BERT in subsequent processing.

ELMo (*Peters et al., 2018*) uses two stages to obtain the word vector. The first stage is to train a deep bi-directional language model (biLSTM) with a large number of text corpus; the second stage is to extract the internal state of each layer of the network

corresponding to the word from the pre-trained network, and transforms it into the word vector through functions. The structure of ELMo model has been shown in Fig. 1.

Firstly, the maximum length of the sentence is cut to 14, and the sentences with less than 14 words are supplemented to 14 by zero filling. Each word is transformed into an initial word vector of 50 dimensions, that is, the initial word vector $y_k^{LM} \in \mathbb{R}^{50}$, assuming that the number of layers of the bi-directional language model is $L = 2$, the number of hidden layer nodes is $H_{dim}$, and the output dimension is $output_{dim} \in \mathbb{R}^d$, then $ELMO^{task} \in \mathbb{R}^{2d}$, the output text feature $Y \in \mathbb{R}^{n \times 2d}$.

## Multi-head attention mechanism

As mentioned in the introduction, the introduction of attention mechanism helps the neural network to improve the prediction accuracy and reduce the computational complexity. VQA tasks need to process multi-modal data such as images and texts, which requires more efficient computation than tasks that only need to process single-modal data. At the same time, the input image and question text have a high correlation, so the interaction betwen the data of the two modals also has a significant impact on the accuracy of the results. For the above two requirements, N-KBSN uses the multi-head attention (MA) mechanism of Transformer (*Vaswani et al., 2017*) to realize the self attention of image (V-SA), question text self attention (Q-SA) and image attention (GA) guided by questions (*Yu et al., 2019*).

The essence of attention mechanism is to find a way to assign appropriate weight to the existing information and improve the accuracy of the output. According to the multi-head attention (MA) mechanism of Transformer (*Vaswani et al., 2017*), Attention function can be described as mapping query to some key-value pairs and obtaining the output. Suppose the query matrix $Q = \{q_1, q_{2,....}q_m\}$, where the query vector $q_i \in \mathbb{R}^{1 \times d_q}$; the key matrix $Q = \{k_1, k_{2,....}k_n\}k_j \in \mathbb{R}^{1 \times d_k}$, value matrix $Q = \{v_1, v_{2,....}v_n\}$, where the value vector $v_i \in \mathbb{R}^{1 \times d_v}$, then the attention feature can be obtained by weighting the value matrix, and the weight can be obtained by the query matrix and key matrix (*Vaswani et al., 2017*).

In order to further improve the expression ability of attention features, multi-head attention mechanism (*Vaswani et al., 2017*) is introduced. The realization of multi-head attention mechanism is to input Q, K, V into $h$ linear layers with different weights. Finally, $h$ attention features are stitched together and the attention features of the expected dimension are obtained through a linear layer.

Based on the idea of multi-head attention mechanism, this article uses three kinds of attention features: self attention of picture (V-SA), self attention of question text (Q-SA), and image attention guided by question (GA). Assuming that the text word vector matrix is Y and the image feature map is X, then when calculating V-SA, Q = K = V = X, that is, the output image feature is SA = MA (X, X, X); when calculating Q-SA, Q = K = V = Y, that is, the output text feature is SA = MA (Y, Y, Y); when calculating the guided attention feature, Q = Y is the word vector matrix, K = V = X is the image feature matrix, and the word vector and image feature vector have the same dimension, that is, the output image feature guided by the question is GA = MA (Y, X, X).

A Modular Co-Attention (MCA) is composed of three kinds of attention combinations (*Yu et al., 2019*) which takes the the original image feature and text feature as input, and the output is the image and text feature through attention mechanism.

In order to improve the use of deep attention mechanism to extract higher-level features, MCAN article (*Yu et al., 2019*) proposed Encoder-Decoder and Stacking which are two ways of cascading MCA layer. Among them, stacking takes the output of the upper layer as the input of the next layer, and Encoder-Decoder takes the question self attention feature of the last layer as the query matrix of each layer.

According to the performance of the two cascading methods in multiple tasks, N-KBSN model uses the cascade mode of Encoder-Decode.

## Experiment

Our experiment builds a series of VQA models with different network structures or parameter settings, and train and evaluate the models using the general open question-answering data set VQA2.0 (*Goyal et al., 2017*). The purpose of the experiment is to compare the influence of dynamic word vector and static word vector on the accuracy of results, and find the optimal N-KBSN model by a large number of experiments. The whole code is implemented in Python, with pytorch as the machine learning platform, and a computer with 32G memory and GPU is used.

### Experimental setup

1. Data set. In this experiment, VQA2.0 dataset was used to train the model. The data set is divided into three data subsets: train/val/test, which contains 80,000 images + 444,000 question and answer pairs, 40,000 images + 214,000 question and answer pairs, 80,000 images + 448,000 question and answer pairs. The answers include Yes or No, quantity and others. The pictures are real scenes extracted from MS-COCO dataset. In addition, according to the images which are both in VQA2.0 and Visual Genome, 490,000 question and answer pairs are extracted from Visual Genome to enhance the training set.

2. Evaluation method. In order to train and test the open questions, the VQA2.0 dataset designs the questions manually. Each picture has three questions proposed by human beings. The answers are all open and the evaluation method of these answers also introduces artificial evaluation mechanism: for the same open question, ten people answer it separately. If three or more subjects provide the same answer, the answer is regarded as the correct answer. Therefore, the text uses accuracy as the evaluation parameter, including overall accuracy and sub-item accuracy. According to the type of answer, the accuracy of sub-items can be divided into Yes or No, count and others.

3. Parameter setting of fixed module. The purpose of our experiemnt is to compare the effect of different word vector embedding methods on the accuracy of the model, so other parts of the model should keep the same parameter settings. Specifically, for the image feature extraction module, the number of candidate image regions of Fast R-CNN is $m = 100$, and the feature dimension of a single image area is $x_i = 2,048$,

so the single image feature $X \in \mathbb{R}^{100 \times 2,048}$. The hidden layer dimension of MA attention in the SA and GA module is $d = 512$, the number of heads $h = 8$, that is, the hidden layer dimension of each $head_h = d/h = 64$, and the number of MCA layers $L = 6$. We select words or phrases with more than eight occurrences from all the answers, and construct an answer dictionary with the size of $n = 3,129$, that is, the number of classified categories is 3,129.

The activation function uses ReLU. The parameters of Adam optimizer are $\beta_1 = 0.9$, $\beta_2 = 0.98$ and the learning rate is $\min(2.5te^{-5}, 1e^{-4})$, where $t$ is the number of trained epochs. Starting from the 10th generation, the learning rate of every two generations decreases to 1/5 of the current. The number of batch samples is 32, and the training epoch is 13.

With the rest of the fixed model unchanged, this experiment will construct models using different text characterization methods to evaluate the impact of dynamic and static word vectors on the accuracy of the results. And the technical details of our model is shown in Fig. 2.

### Model based on static word vector

In the experiment, the representative and widely used pre-trained word2vec and Glove word vectors are selected in the experiment, and a single layer LSTM network is cascaded to convert the feature dimension into 512, which is convenient for the fusion with image features subsequently. The word2vec word vector uses the word vector obtained from word2vec model which has been trained on Google News. And the Google News contains 100 billion words and phrases. Word2vec word vector is composed of a vector representing three million words and phrases.

The Glove word vector is obtained by being trained on the corpus of Wikipedia and twitter, which contains 2 million vectors with the dimension of 300. In order to use the static word vector, the question text of the input sample is cut into a 14 word sequence, and then the static vector of each word is obtained by using the look-up table. If there is no pre-traind word vector in the data set, the word vector is initialized as zero vector.

### Model based on dynamic word vector

Different from the static vector configuration, in order to obtain the dynamic word vector, the pre-trained deep bidirectional LSTM network (biLSTM) is embedded into the model, and the weight parameters and scale parameters of ELMo word vector are obtained by training. In order to further explore the best ELMo parameters, three pre-trained ELMo models with different parameters are selected in this experiment, which are $ELMO_S/ELMO_m/ELMO_l$. The parameters, hidden layer size, output size, ELMo size and LSTM size are shown in Table 2. As shown in the table, the main differences of the three different ELMo models lie in model depth and the dimension of word vector. Theoretically, the deeper network depth, the greater capacity. And the higher dimensional word vector can contain more semantic information. In the same way, the question text is cut into the sequence containing 14 words, and the whole word sequence is taken as input. Through two-layer biLSTM network, ELMo word vector containing the context is

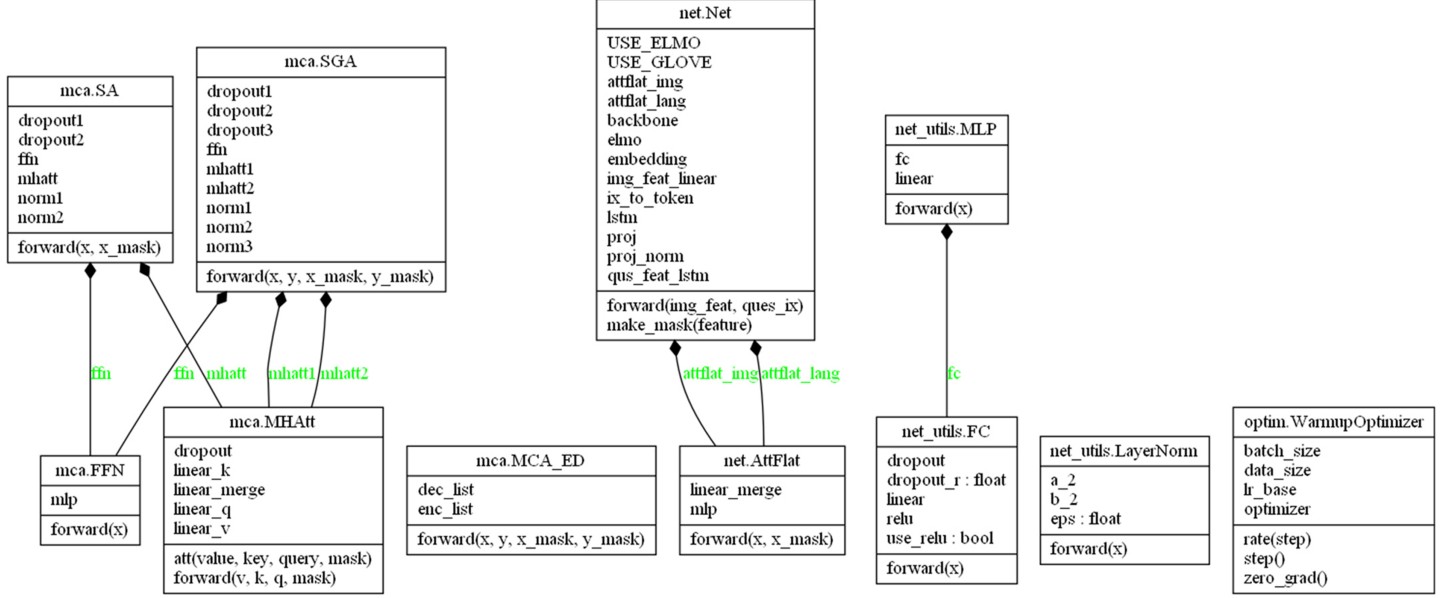

**Figure 2 UML.** Technical details of N-KBSN model based on dynamic word vector. net. Net: image feature, text feature extraction; mca. SA: the self-attention mechanism; mca. SGA: the mutual attention mechanism of text and image; net_utils. MLP: multi-layer perceptron; net_utils. FC: fully connected layer; net_utils. LayerNorm: normalized layer; optim. Wamupoptimizer: optimizer with warmup mechanism; mca. MHAtt: multi-head attention mechanism; mca. MCA_ED: MCA layers cascaded by Encoder-Decoder; net. AttFlat: Flatten the sequence; mca.FFN: feed forward nets.

**Table 2 Parameter configuration of ELMO.**

| Model | Parameters (M) | LSTM Size | Output Size | ELMo Size |
|-------|----------------|-----------|-------------|-----------|
| $ELMo_s$ | 13.6 | 1,024 | 128 | 256 |
| $ELMo_m$ | 28.0 | 2,048 | 256 | 512 |
| $ELMo_l$ | 93.6 | 4,096 | 512 | 1,024 |

**Table 3 Statistics information of word2vec, glove and three Elmo models.**

| Name | Pre-training corpus (size) | Word vector dimension | Number of word vectors |
|------|----------------------------|----------------------|------------------------|
| word2vec | Google News (100 billion words) | 300 | 3 million |
| Glove | Wikipedia 2014 + Gigaword 5 (Six billion words) | 300 | 400 thousand |
| $ELMo_s$ | | 256 | |
| $ELMo_m$ | WMT 2011 (800 million words) | 512 | / |
| $ELMo_l$ | | 1,024 | |

obtained. Then the obtained ELMo word vector is unified into 512 dimensions through a LSTM network, and the fused feature $Z \in \mathbb{R}^{512}$.

Statistics of several word vectors are shown in Table 3. Because ELMo model uses character-level coding, even for words that do not exist in the corpus, the initial word vector can still be obtained, and then the ELMo word vector can be obtained. Therefore, the number of ELMo word vectors is theoretically infinite.

**Table 4 The results of different text features characterization model in the Val dataset.**

| Model | Accuracy | Yes or No | Count | Others |
|-------|----------|-----------|-------|--------|
| *baseline (random)* | 62.34 | 78.77 | 41.92 | 55.27 |
| *2017-winner(glove)* | 63.22 | 80.07 | 42.87 | 55.81 |
| *baseline (w2v)* | 64.37 | 81.89 | 44.51 | 56.31 |
| *baseline(glove)* | 66.73 | 84.56 | 49.52 | 57.72 |
| *2019-winner(glove)* | 67.22 | 84.80 | 49.30 | 58.60 |
| *N-KBSN(s)* | 67.27 | 84.76 | 49.31 | 58.73 |
| *N-KBSN(m)* | 67.55 | 85.03 | 49.62 | 59.01 |
| *N-KBSN(l)* | 67.72 | 85.22 | 49.63 | 59.20 |

In order to facilitate the analysis of experimental results, the models with different text features are represented as *baseline(w2v), baseline (glove), N-KBSN(s), N-KBSN(m), N-KBSN(l)*.

All in all, in this experiment, VQA2.0 dataset is used to train and evaluate the six models: *baseline(random), baseline (w2v), baseline (glove), N-KBSN(s), N-KBSN(m), N-KBSN(l)*. Among which, the word vector of *baseline(random)* was randomly initialized and cascaded with LSTM network.

At the same time, the model comparison also includes the 2017—*winner(glove)* (*Teney et al., 2018*) and 2019—*winner(glove)* (*Yu et al., 2019*) model that has won the VQA challenge in the past years.

# RESULTS OF FINDING

## Results of VQA comparison experiment

The experimental results of *baseline(random), baseline (w2v), baseline (glove), N-KBSN(s), N-KBSN(m), N-KBSN(l)* on the val set are shown in Table 4.

## Comparison about N-KBSN(m) model and *baseline (glove) model*

As shown in Table 4, the experimental results of *baseline (glove) is better than baseline (w2v)*, and *N-KBSN* model using ELMo dynamic vector are better than *baseline(glove)*. Specifically, the ELMo model parameters of N-KBSN (l) are more than three times of that of N-KBSN (m), but the accuracy rate is not significantly improved. Consequently, we would conduct comparison experiments between *N-KBSN(m)* and *baseline (glove)* to find out where the difference lie in.

In order to quantitatively analyze the difference between *N-KBSN(m)* and *baseline (glove)*, the VQA2.0 dataset is randomly sampled to form training subsets with the size of 10%, 30%, 50%, 70%, 90% and 100% of the original size respectively, and the two models are trained according to the same parameter configuration in previous experiments. The overall accuracy of the two models on the val dataset is shown in Table 5, and the diagram is shown in Fig. 3.

In order to qualitatively analyze the difference between the results of *N-KBSN(m)* and *baseline (glove)*, we compare and analyze the prediction results of the two models

**Table 5 Statistics of experimental results of *N-KBSN(m)*and *baseline (glove)* in different training subsets.**

| Accuracy | *baseline (glove)* | *N-KBSN(m)* | Difference |
|---|---|---|---|
| Training subset (10%) | 54.65 | 55.03 | 0.38 |
| Training subset (30%) | 60.30 | 60.81 | 0.51 |
| Training subset (50%) | 62.87 | 63.40 | 0.53 |
| Training subset (70%) | 64.12 | 64.72 | 0.60 |
| Training subset (90%) | 65.98 | 66.69 | 0.71 |
| Training subset (100%) | 66.73 | 67.55 | 0.82 |

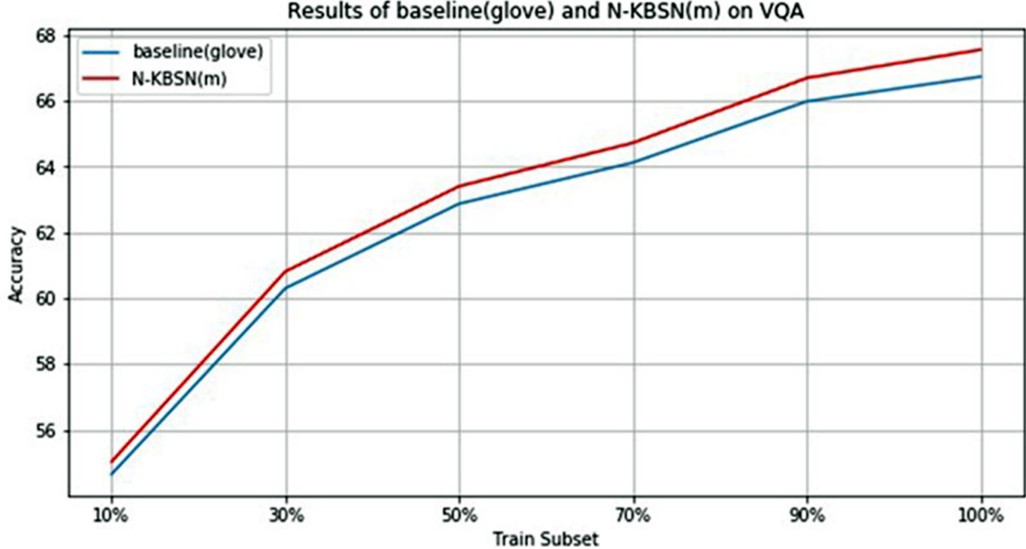

**Figure 3 Statistics chart of experimental results of *N-KBSN(m)* and *baseline (glove)* in different training subsets.**

in the val dataset, and uses *Res(ELMo)* and *Res(glove)* to represent their result sets respectively. The size of *Res(ELMo)* and *Res(glove)* is the number of questions in the val set: 214354. The number of questions with different answers given by the two models is 52990, accounting for about 1/5 of the total. Among the different answers given, *Res(ELMo)* answered correctly and *Res(glove)* answered wrongly accounted for 27.8%, *Res(glove)* answered correctly while *Res(ELMo)* answered wrong accounted for 26.4%, the two both answered wrongly accounted for 45.8%, and the two both answered correctly accounted for 0%, as shown in Table 6.

In order to further qualitatively analyze the possible reasons why N-KBSN(m) is better than *baseline (glove)*, we selected some samples from the answers, as shown in Fig. 4.

## Comparison between existing models

As shown in Table 4, *N-KBSN(l)* performs best on the val set. The results of the model and *2017—winner (glove)* and *2019—winner (glove)* on Test-dev and Test-std are shown in Table 7.

**Table 6 The difference between the answers of *Res(elmo)*and *Res (glove)* in the validation set.**

The number of questions with different answers (proportion) 52,990 (24.7%)

Only *Res(ElMo)* answered correctly (proportion) 14,711 (27.8%)

Only *Res(glove)* answered correctly (proportion) 13,991 (26.4%)

Both answered wrongly (proportion) 24,288 (45.8%)

Both answered correctly (proportion) 0 (0%)

| Q: | What color is the shower curtain? | How many elderly people are in this picture? | What fashion accessory is sitting on top of the white collared shirt? |
|---|---|---|---|
| Res(elmo): | Blue ✓ | 0 ✓ | Tie ✓ |
| Res(glove): | White | 5 | Hat |

**Figure 4 Sample examples of validation sets.**

**Table 7 Comparison of the results of *N-KBSN(l)* model and other models on test set.**

| Model | Test-dev | | | | Test-std All |
|---|---|---|---|---|---|
| | All | Y/N | Num | Other | |
| 2017—*winner (glove)* | 65.32 | 81.82 | 44.21 | 56.05 | 65.67 |
| 2019—*winner (glove)* | 70.63 | 86.82 | 53.26 | 60.72 | 70.90 |
| *N-KBSN(l)* | 71.14 | 87.13 | 54.05 | 60.90 | 71.23 |

# DISCUSSION

## Discussion about VQA experiment results

In the real language environment, the complex characteristics of word use (e.g., syntax and semantics), and how these uses vary across linguistic contexts cannot be effectively expressed by static word vectors, so there may be semantic and grammatical deviations. So we introduce dynamic word vector into the VQA models, and do experiments on the *baseline(random), baseline (w2v), baseline (glove), N-KBSN(s), N-KBSN(m), N-KBSN(l)* models. As shown in the Table 4, in terms of the accuracy of the answer type, the experimental results of all models are as follows: Y/N > others > count, and the accuracy difference of any two remains stable in the results of different models. This shows that the accuracy difference of the single type is independent of the model and comes from the characteristics of the question itself and the data set. For example, the expected accuracy rate of the sample with the answer type of Yes or No is 50%, while that of the answer type of count is very low. The difference of difficulty between the two types of answers leads to that the accuracy of the Yes or No type is always higher than that of the count type.

The first five models all use static word vectors as text features. *baseline (random)* uses randomly initialized text features, which contains less semantic and grammatical information than the pre-trained word vector, so its accuracy is the lowest. 2017—*winner*

*(glove)* has a lower accuracy than the baseline model because of its simpler attention mechanism. Comparing the results of *baseline(w2v)* and *baseline(glove)*, we can see that even if the corpus of word2vec word vector is nearly 20 times as large as that of glove word vector, the model based on glove still takes the lead in all aspects, even if other parts are the same. As the article (*Pennington, Socher & Manning, 2014*) said, the glove word vector uses co-occurrence matrix. Compared with word2vec word vector, which only uses local context information, it introduces the global information of corpus and improves the representation ability. Therefore, even if a larger corpus is used, the representation ability of word2vec word vector is still worse than that of glove word vector.

The last three models are the N-KBSN model proposed in our experiment. From the results, we can see that compared with *baseline(glove)*, the accuracy of each item is significantly improved, which proves that the dynamic word vector can improve the text feature representation ability of the model to a certain extent, and then improve the overall result accuracy. And the accuracy of the three N-KBSN models are all higher than the 2019—*winner(glove)*.

Compare the ELMo model with three different parameters, it is not difficult to find that with the increase of model depth and feature dimension, the overall accuracy rate is improved, but the improvement range is gradually reduced. Specifically, the parameters of ELMo model of *N-KBSN(l)* are more than three times of that of *N-KBSN(m)*, but the accuracy rate is not significantly improved.

## Discussion about the comparison of N-KBSN(m) model and *baseline (glove)* model

According to the above Fig. 3 and Table 5, with the increase of training data, their accuracy rates monotonically increase, and the rising speed gradually decreases, making the accuracy rate tend to be stable. This shows that increasing the amount of training data can help improve the accuracy, but the improvement will tend to saturation. It is worth noting that only 10% of the training data can get better accuracy.

We want to determine whether the difference of accuracy is statistically significant between the *N-KBSN(m)* and *baseline (glove)*. We use the Wilcoxon signed rank test and the results showed in Fig. 5 indicate $p < 0.05$. Consequently, the difference is statistically significant.

The accuracy of *N-KBSN(m)* is always higher than *baseline (glove)*, which shows that dynamic word vector can improve the accuracy of the model. With the increase of data, the difference of accuracy between the two models gradually increases. This is because the dynamic word vector can effectively represent the performance of polysemous words. Larger training data means that it is more likely to include the use of words in different contexts, while static word vector can not effectively deal with this situation.

We compare and analyze the prediction results of the *N-KBSN(m)* and *baseline (glove)* in the val dataset as shown in Table 5. From the overall answer difference, the number of correct answers of *N-KBSN(m)* is slightly higher than that of *baseline (glove)*. As shown in Fig. 4, *Res(ELMo)* correctly identifies the color of bath curtain, while *Res*

**Test Statistics[a]**

| | difference - NKBSN_M |
|---|---|
| Z | -2.201[b] |
| Asymp. Sig. (2-tailed) | .028 |

a. Wilcoxon Signed Ranks Test

b. Based on positive ranks.

(A)

**Report**

Mean

| baselineGlove | NKBSN_M | difference |
|---|---|---|
| 62.4417 | 63.0333 | .5917 |

(B)

**Figure 5 Statistical results of difference between N-KBSN(m) model and baseline (glove) model.**
(A) Wilcoxon Signed Ranks Test Statistics; (B) Accuracy difference.

**Test Statistics[a]**

| | difference - winnerGlove2017 |
|---|---|
| Z | -2.023[b] |
| Asymp. Sig. (2-tailed) | .043 |

a. Wilcoxon Signed Ranks Test

b. Based on positive ranks.

(A)

**Report**

Mean

| NKBSNL | winnerGlove2017 | difference |
|---|---|---|
| 68.8900 | 62.6140 | 6.2760 |

(B)

**Figure 6 Statistical results of difference between N-KBSN(m) model and 2017winner(glove) model.**
(A) Wilcoxon Signed Ranks Test Statistics; (B) Accuracy difference.

*(glove)* answers white. The possible reason is that the model mistakenly identifies the color of shower instead of shower curtain. Similarly, *Res(glove)* wrongly counts the phrase people while *Res(ELMo)* correctly counts the phrase elderly people. Figure 4 shows that *Res (ELMo)* performs better in recognizing phrases in long sentences rather than just words.

## Discussion about comparison with existing models

The result on the test set shown in Table 7 indicates that the *N-KBSN(l)* proposed in our experiment is better than the other 2017—*winner(glove)* and 2019—*winner (glove)* models in each index, and the results are consistent with the results on the val set, which proves that the improved text characterization method can improve the prediction accuracy.

We compared the difference between *N-KBSN(l)* and 2017—*winner (glove)*, and the difference between *N-KBSN(l)* and 2019—*winner (glove)*. We want to determine whether the difference of accuracy is statistically significant between the visual question answering model based on dynamic word vector and the visual question answering model based

## Test Statistics[a]

| | difference -<br>winnerGlove2<br>019 |
|---|---|
| Z | -2.023[b] |
| Asymp. Sig. (2-tailed) | .043 |

a. Wilcoxon Signed Ranks Test

b. Based on positive ranks.

(A)

## Report

Mean

| | winnerGlove2 | |
|---|---|---|
| NKBSNL | 019 | difference |
| 68.8900 | 68.4660 | .4240 |

(B)

**Figure 7 Statistical results of difference between N-KBSN(l) model and 2019winner (glove) model.**
(A) Wilcoxon Signed Ranks Test Statistics; (B) Accuracy difference.

on static word vector. We use Wilcoxon signed rank test to test the statistical significance, and the statistical results are shown in Figs. 6 and 7. According to the statistical results, $P < 0.05$ for the difference between *N-KBSN(l)* and 2017—*winner (glove)*, and the difference between *N-KBSN(l)* and 2019—*winner (glove)*. The results indicate that the difference is statistically significant.

We have dicsussed the current joint-embedding VQA models in detail in the Introduction secction and found that all existing models use static word vectors for text characterization ignoring the fact that the same word may represent different meanings in different contexts, and may also be used as different grammatical components. Taking into the problem brought by the static word vectors into account, our experiment constructs a joint embedding model based on dynamic word vector—N-KBSN model and proves that it can improve the accuracy of the results through the comparison shown in the "Results of VQA Comparison Experiment" and "Comparison Between Existing Models".

We simply propose the thought that the introduction of dynamic word vector into VQA system can improve the accuracy and do a simple experiment to find out the results, so the findings are preliminary and can be improved a lot in the future. For example, since the image feature characterization method is not the research content of this article, we directly use the pre-trained Faster R-CNN network to extract image features. The design of parameters also migrates the champion model of VQA challenge in 2017, and only fine tune the parameters in the training. If we want to further improve the accuracy of the model, on the one hand, we can search the network to obtain the best combination of hyperparameters, on the other hand, we can use better image recognition models such as Mask R-CNN.

## CONCLUSION

This article constructs and tests a series of VQA models based on static word vector and dynamic word vector of different parameter configurations. The experimental results show

that the *N-KBSN(l)* proposed in our experiment has better performance than the other *2017—winner (glove)* and *2019—winner (glove)* models. Consequently, the introduction of dynamic word vector into the VQA model improves the accuracy of the results to some extent.

### Funding

This work was jointly supported by the Sichuan Science and Technology Program (2019YJ0189) and the Fundamental Research Funds for the Central Universities (ZYGX2019J059). The funders had no role in study design, data collection and analysis, decision to publish, or preparation of the manuscript.

### Grant Disclosures

The following grant information was disclosed by the authors:
Sichuan Science and Technology Program: 2019YJ0189.
Fundamental Research Funds for the Central Universities: ZYGX2019J059.

### Competing Interests

The authors declare that they have no competing interests.

### Author Contributions

- Zhiyang Ma conceived and designed the experiments, performed the experiments, analyzed the data, performed the computation work, prepared figures and/or tables, authored or reviewed drafts of the paper, and approved the final draft.
- Wenfeng Zheng conceived and designed the experiments, performed the experiments, prepared figures and/or tables, and approved the final draft.
- Xiaobing Chen performed the experiments, analyzed the data, performed the computation work, prepared figures and/or tables, authored or reviewed drafts of the paper, and approved the final draft.
- Lirong Yin performed the experiments, analyzed the data, performed the computation work, prepared figures and/or tables, and approved the final draft.

### Data Availability

Code is available at GitHub: https://github.com/summermzy/NKBSN.
It is also available at Figshare:
Ma, Zhiyang (2020): elmo_2x4096_512_2048cnn_2xhighway_weights.hdf5. figshare. Dataset. DOI 10.6084/m9.figshare.13238969.v1.

### Supplemental Information

Supplemental information for this article can be found online at http://dx.doi.org/10.7717/peerj-cs.353#supplemental-information.

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
