# Peer review of "Joint embedding VQA model based on dynamic word vector"

_PeerJ Computer Science, doi:10.7717/peerj-cs.353_

## Round 0.1 · original submission · Major Revisions

The manuscript must be revised carefully following the comments and suggestions of the reviewers. Additionally, the authors must supplement the article with the statistical analysis of the results section in order to demonstrate that the results achieved by the authors are statistically significantly better than the baseline. In particular, the use of appropriate statistical measures (beyond commonly used descriptive statistics), correction for multiple testing, and reporting of relevant figures such as the statistic, n, and p-values/confidence intervals as needed is required.

Reviewer 1 ·

Basic reporting

Literature is ok.
The hypothesis is ok.
The experimental section should be extended to more comparison and deeper statistical analysis.

Experimental design

More comparison and more statistical analysis is needed.

Validity of the findings

Use more public available datasets to conduct more tests and comparisons.

Additional comments

In general, the paper is very good written, but some issues must be improved:
1) Equations should have some interpunction in the end. A word where after equations should be written in small letters.
2) Colors should be standardized for all figures.
3) Charts are too small.

Reviewer 2 ·

Basic reporting

Introduction should also mention application in disabilities, e.g.:https://doi.org/10.1155/2016/3054258; gaming: https://doi.org/10.1109/TG.2019.2896017; chatbots: https://doi.org/10.1007/978-3-030-17705-8_8

Add sample, illustratory outputs to figures 1 and 2

Remove Figure 3 as its also part of Figure 1

Consider adding Figure 4 as part of Figure 1

Basic implementation is explained (generic scheme was provided in figures 1-5), but paper lacks the minimum technical details on the factual implementation. Supplement the text with a UML based use case diagram and an activity diagram (logical sequence of the code) explaining the implementation. You'll find instructions (if needed) here:
https://www.tutorialspoint.com/uml/uml_use_case_diagram.htm
https://www.tutorialspoint.com/uml/uml_activity_diagram.htm
Add all parameters, discuss optimizer parts.

Experimental design

Authors present an approach based on FasterCNN for image classification and Elmo for text processing, which is quite known solution, therefore I suggest the authors highlight or demonstrate other contributions in the algorithm part. Novelty aspect was not reflected in the abstract and introduction too.
The experiments should be contextualized better (the reader should not be left to assume that they will get their own conclusions). The experiments should be described more clearly (e.g. set up and carry out process, results in raw format, etc.). Add a statistical reliability analysis. Discuss the impact on precision vs resource intensiveness.

Validity of the findings

Results are comparable to those of other authors: https://allennlp.org/elmo
More vigorous discussion should be added clarifying what has been achieved and whats the difference (if at all)... Note, that a different application or a combination of standard methods is still not considered as satisfactory merit for basis of novelty required for scientific publication, so I'd suggest carefully revising such parts in the paper.

Reviewer 3 ·

Basic reporting

This paper provides a framework for VQA model using dynamic word vector.
The framework consists of several well-explored components.
The paper is well written.
The font of the equations needs to be double checked.

Experimental design

Extensive experiments are conducted, although the noverty is limited.

Validity of the findings

The research appears to be sound, the results are exciting for their potential as a research tool, the study will be of interest to PeerJ readers.

Additional comments

I believe that the paper merits publication in PeerJ. However, I would suggest the author to use Latex for the manuscript editing, as it is quite difficult to follow the equations in current status.

---

## Round 0.2 · Major Revisions

Please revise carefully following the comments of the reviewers. Specifically, please improve the statistical analysis part of the manuscript.

Reviewer 2 ·

Basic reporting

UMLs must be added to provide technical explanation of the working principle as was originally requested.

Experimental design

Requested explanation was provided.

Validity of the findings

Statistical analysis still needs to be more detailed. Reliability of results is still not clear.

Attached code does not work. Add a full working project! Were some parts cut from the model?

Reviewer 4 ·

Basic reporting

English has to be improved (the text contains spelling errors).
Literature review is ok. References to formulas, Figures (not invented by authors must be referenced).
The article structure is ok.
The contribution (the novelty) of the paper is not emphasized.

Experimental design

Research is within Aims and Scope of the journal.
Research questions are well defined.
Methods are not invented by the authors. They are formally (mathematically) described.

Validity of the findings

Novelty must be emphasized.
The results are close to the baseline. The authors must evaluate if differences are statistically significant before making the conclusions that their method is better compared to the baseline models.

Additional comments

You have considered remarks of the other reviewers and already improved your description. The work is interesting, the topic is important. Unfortunately, I still cannot say that the description is ready for publishing. There are my comments:
1. “…but all existing models use static word vectors for text characterization. However, in the real language environment, the same word may represent different meanings in different contexts, and may also be used as different grammatical components.” (line 18-21). E.g., Elmo, BERT embeddings successfully tackle this problem: they solve ambiguity problems by vectorizing each word depending on its context. Thus, not all existing models use only “static word vectors”.
2. You state that your model based on ELMo vectorization outperforms glove. However, it is not very surprising. ELMo considers the word order and uses LSTMs. It is based on more sophisticated technique compared to glove.
3. You are using Faster R-CNN, ELMo embeddings and multi-head attention mechanism. All these techniques are not novel. You must emphasize the contribution of your work.
4. You make the overview of different research works in the field; however, it is not clear how your work differs from the other.
5. The formal description takes 5-6 pages. I do not have anything against math, but these formulas are not invented by you. You must add references, where needed; explain what parameter values have been set in your experiments and why.
6. What is the point of presenting Figure 1-Figure 5? Aren’t they presenting the basic architectures? If I am wrong, you must emphasize the differences; otherwise, please, add references, where needed.
7. The results in Figure 6 of the baseline and N-KBSN are very similar (~1% difference) it arises the doubts if differences are statistically significant. This also questions the validity of your Conclusions.
8. The discussion section is vague, especially explaining the results related with the image data (Section 5.3).
9. The grammatical errors (e.g., table8, secction); not unified writing style (ELMo, Elmo; Bert, BERT; biLSTM, bilstm) must be considered as well.

---

## Round 0.3 · Minor Revisions

The authors do not use the terms "statistics", "statistical" in the correct way. The statistical reliability of the results must be evaluated in terms of standard deviation, confidence limits or similar. Statistical significance must be supported by statistical testing such as signed ranking test.

Reviewer 4 ·

Basic reporting

no comment

Experimental design

no comment

Validity of the findings

no comment

Additional comments

The authors partially considered my previous comments.

However, I still have not got the answer about if differences between their results are statistically significant. Your statement "Because the training data used for experiment is limited, the differences are statistically significant" is false: usually having the small dataset is a risk that differences even in several percentage points do not guarantee the statistical significance.

---

## Round 0.4 · Major Revisions

The authors did not provide the answer to my previous editorial comments and did not revise the article accordingly, so I am once again asking the authors to address these issues seriously:

The authors do not use the terms "statistics", "statistical" in the correct way. Figure 7 and Table 4 use the term "statistical" incorrectly. The statistical reliability of the results must be evaluated in terms of standard deviation, confidence limits or similar. Statistical significance must be supported by statistical testing such as a signed ranking test.

The authors claim multiple times that the accuracy is or is not significantly improved with respect to the comparison baselines, however, such claims must be supported by the results of statistical testing. The statistical analysis section must be included in the article. to accommodate the statistical tests and discuss the results.

Also, the answer to the comments of Review 4 is not satisfactory, as the authors did not include the discussion on the statistical significance of the results in the article.

---

## Round 0.5 · accepted · Accept

The authors have addressed the comments sufficiently. However, the authors should perform spellchecking and carefully check the correctness of bibliographic data in references before submitting the final version for publication.